# Minimizing Deformations in High-Temperature Vacuum Carburizing

**DOI:** 10.3390/ma16247630

**Published:** 2023-12-13

**Authors:** Radomir Piotr Atraszkiewicz, Konrad Dybowski

**Affiliations:** Institute of Materials Science and Engineering, Lodz University of Technology, Stefanowskiego 1/15, 90-924 Lodz, Poland; konrad.dybowski@p.lodz.pl

**Keywords:** effusion, vacuum carburizing, deformations

## Abstract

This article presents the results of a study on reducing deformations resulting from high-temperature vacuum carburizing and post-carburizing heat treatment. The idea was to increase the strength of steel at elevated temperatures by pre-carburizing at heat-up to the process temperature (SC—stage carburizing). It has been shown that the use of carburizing in stages from a lower temperature to the target temperature, compared to traditional vacuum carburizing at a constant temperature (Constant-Temperature Carburizing—CTC), has a significant impact on the chemical and phase composition of the technological layer, surface after the process and, consequently, on its mechanical properties. It was shown that the retained austenite content after stage carburizing was reduced by approximately 45%, as was the thickness of the gear teeth measured at the pitch diameter. Additionally, uniform stress distribution was demonstrated for the SC process. Carbon saturation of austenite increases the yield strength, and therefore the dimensional stability of steel heat-treated at elevated temperatures also improves, which effectively permits high-temperature treatment of critical steel parts such as, for example, gear wheels, for which high dimensional accuracy is required.

## 1. Introduction

Carburizing is one of the major surface treatments of steel. It leads to an increase in case hardness and strength while allowing the core of the steel workpiece to retain its plasticity. The most recent variation of that process is carburizing carried out at reduced-pressure atmospheres, which is known as low-pressure or vacuum carburizing. The concept of carburizing at reduced pressure was created in the mid-1970s [1,2], but at the beginning it did not gain acceptance in the industry due to the lack of recognition of the physical chemistry of the process and, consequently, the lack of repeatability and effective control systems. Only the second half of the 1990s saw progress in vacuum carburizing technology, resulting in the appearance of modern technologies and devices [3,4,5]. The technology has a number of advantages over carburization at endothermic atmospheres, including the possibility of performing the process at temperatures exceeding 1000 °C [6,7,8], and is now replacing the latter method that was commonly used before. 

Achieving the required depth of the carburized layer in the shortest possible time is the key requirement in terms of efficiency of the carburizing process. Process temperature is the most crucial factor that determines the rate of carburization. An increase in the carburizing temperature intensifies the thermal decomposition of hydrocarbons in the carburizing atmosphere, which increases the rate of diffusion and solubility of carbon in austenite [7,8]. However, the rise in the carburizing temperature may cause excessive post-heat-treatment deformations [9,10,11,12,13]. Carburizing is mostly used on critical parts for which high dimensional accuracy is required. Deformations are a side effect of thermo-chemical treatment at elevated temperatures [14,15,16,17,18,19]. The extent of deformations resulting from carburizing and subsequent heat treatment is a function of thermal/plastic and structural deformations. Intensification of carburizing affected by raising the process temperature may result in an increase in thermal/plastic deformations only if post-carburizing heat treatment is carried out at the same process parameters, regardless of the carburizing temperature, and if the austenite primary grain size remains comparatively constant. In that case, the difference in deformation between low-temperature and high-temperature carburizing will be limited to creep experienced during the carburizing cycle [20,21,22,23].

The increase in the process temperature, which has a direct effect on the intensification of the carburizing process, is constricted by allowable deformation tolerance for a given heat-treated part. Correction of the part geometry due to the deformations involves grinding of the parts after the process, which causes buildup of adverse residual stresses in the case/surface layer [23,24,25]. Their values depend on the depth and parameters of the grinding. In extreme cases, where deformations are extensive, geometry correction may be difficult, if not impossible. Hence, considering the efficiency of the carburizing process, it is crucial to control this phenomenon. One of the principal solutions applied in this regard is deformation control, in which tools are used to influence the shape and dimensional stabilization. However, this solution cannot be successfully applied to low-pressure carburizing, where vacuum heat treatment occurs immediately after carbon diffusion. This method significantly reduces the overall time of the treatment, yet it prohibits the use of tools that minimize deformations. Therefore, it is necessary to develop other techniques to control vacuum carburizing deformations.

This article presents a novel solution and reports the results of its tests. Increased hardness of steel at high temperatures is achieved by carburizing with heat until the process temperature is reached. The basis of this concept lies in the fact that the yield strength of steel can be increased by raising carbon content. For example, if, for the material that is the subject of this publication, i.e., the 16MnCr5 alloy, we analyze the influence of temperature and carbon content on the value of the yield strength we will observe that, at a temperature of 800 °C, the yield strength at the carbon content 0.16 wt.% is 72 MPa. Increasing the carbon content to 1 wt.% C causes this limit to increase to 235 MPa. This tendency is repeatable for higher temperatures, e.g., at 1000 °C, and at carbon content 0.165 wt.% C, the yield strength is 66.8 MPa, and increasing to 1 wt.% C causes this limit to increase to 228 MPa. This means that increasing the carbon content in the alloy at higher temperatures results in a proportional increase in the yield strength value by up to three times. Hence, the dimensional stability of high-carbon austenite parts should be higher than that of low-carbon austenite. This article presents the results of experiments that confirm the above claim.

## 2. Materials and Methods

Gears were made of 16MnCr5 (Table 1). After machining, the gears were vacuum annealed at a temperature of 860 °C to normalize the structure and relieve internal stresses. Geometry parameters of the tested gears are presented in Table 2.

The gears were then low-pressure carburized [26] at a temperature 1000 °C (CTC—Constant-Temperature Carburizing) and low-pressure carburized at the same temperature but with pre-carburizing (SC—Stage Carburizing). One-minute boost and ten-minute diffusion cycles were used for the SC process during an isothermal hold at temperatures of 860 °C, 900 °C, 950 °C and 1000 °C. In the first and second case, the effective case depth, 0.7 mm, was set at 0.4 weight % of C and surface carbon concentration, Cp, at 0.8 weight % of C. To obtain the desired carbon profile, process parameters for each experimental case were determined based on a SimVaCPlus^®^ 3.0 simulation [27].

Low-pressure carburizing was carried out in a VPT-4022/24IQN HPGQ universal vacuum carburizing furnace (Seco/Warwick Group, Swiebodzin, Poland). The atmosphere consisted of a mixture of aliphatic compounds of ethylene and acetylene diluted with hydrogen [21]. Heating to the process temperature was performed at the same rate for both cases (Vn = 10 °C/min). The pressure in the chamber during the carburizing cycle was Pp = 300 ÷ 800 Pa (pressure pulsation), and Pw = 10 Pa during the diffusion cycle. After carburizing, high-pressure gas quenching was realized with an absolute pressure of 1.4 MPa. In order to fully observe the phenomena resulting from carburizing and hardening, low post-process tempering was not used.

### 2.1. Chemical Composition by GDOES

The chemical components concentration gradients were performed by means of the quantitative depth profiling (QDP) method using optical emission spectrometry with glow discharge (GDOES) on a LECO GDS-850A device (LECO Corporation, St. Joseph, MI, USA).

The measurement parameters for the measured materials in the case of the above-mentioned methods were voltage Ug = 700 V, intensity Ig = 30 mA, and pressure pg = 1.33 kPa. Measurements were taken with a frequency of 50 Hz. The measurement accuracy of the method used is at the level of ±0.015%.

### 2.2. Microstructure and Hardness Assessment

Microstructure studies were carried out using a Keyence VHX-6000 digital microscope (Osaka, Japan) with a VHZ100R lens and VHX-H2M2, ver. 2.08, software. The structure of the surface layers presented in the manuscript was revealed using Beraha II reagent.

Hardness tests were carried out using a KB Prüftechnik GmbH hardness tester, type KB105VZ-FA (Hochdorf-Assenheim, Germany), and KB Prüftechnik Hardwin XL v2.4.15 software. Measurements were made at1 kG load.

### 2.3. Residual Stress and Retained Austenite Content Measurements by XRD

The measurements of residual stresses in the samples were carried out using the sin2ψ X-ray method in ω geometry using a PROTO iXRD apparatus (Proto Manufacturing Inc., Taylor, MI, USA) equipped with two position-sensitive semiconductor detectors. The source of X-radiation was a Cr anode tube emitting characteristic X-radiation with a wavelength of λ = 2.29 A. The change in position (211) of the iron characteristic peak, positioned at an angle of 2θ = 156.4°, was examined. The X-ray elastic constants of ½ S2 = 5.92 1/TPa and S1 = −1.27 1/TPa were measured for calculation purposes. The measurement was carried out for an area limited by a collimator of diameter φ = 2 mm.

To characterize the concentration of retained austenite using XRD, the same diffractometer was used as described above, and four diffraction peaks were collected by the instrument: two for the ferrite/martensite phase (200) and (211) at 106° and 156° 2Theta, respectively, and two for the austenite phase −79° (200) and 128° (220). A comparison of the intensities of the four peaks yields the volume percent concentration of retained austenite in the sample. All calculations and procedures were performed in accordance with ASTM E-975 standard [28].

### 2.4. Geometry Changes Assessment

Measurements of the gear wheels before and after each variant of the carburizing process were taken in order to determine their impact on the extent of deformation. Each time, positioning of the gears in the furnace chamber was performed to exactly the same specification. The following fundamental geometrical parameters of the gears were measured:Tooth thickness;Tooth run-out at pitch diameter;Deformation of the bore diameter.

The measurements were performed with Carl Zeiss (Jena, Germany) gear wheel measuring instrument. The pitch diameter was measured by taking the radius from the axis of rotation to the pitch diameter of each tooth. Tooth thickness was taken at the pitch diameter (circle). These measurements were used to determine deformations for each of the carburizing experiments.

Deformation of the bore diameter, tooth run-out at pitch diameter and tooth thickness was conducted with an electronic gauge of scale division 0.001 mm. The total error of measurement was 0.003 mm.

To demonstrate the significance of the differences in the obtained test results, an ANOVA test (Analysis of Variance) was performed to determine the tooth thickness and bore diameter. In order to use the ANOVA test, first of all, the results were checked to see whether the distribution of values is normal. Next, an ANOVA test was performed assuming significance *p* ≤ 0.05, followed by the Bonferroni test, which showed the significance of differences in values between respective groups of test results.

## 3. Results

### 3.1. Chemical Composition

First, the chemical composition tests checked and compared the carbon profiles obtained during the step carburizing (SC) and Constant-Temperature Carburizing (CTC) processes (Figure 1). The chart also shows simulated carbon profiles after step carburizing at temperatures of 860, 900 and 950 °C.

Figure 1 proves that the SC and CTC processes were carried out correctly, resulting in practically identical carbon profiles. The wheel from the CTC process has slightly higher carbon content on the surface. Both the profile slope and the effective layer thickness are the same. Additionally, the graph shows simulated results of carbon profiles after carburizing stages at temperatures of 860, 900, and 950 °C, conducted during the SC process. Their surface concentration does not exceed 0.5%C, which was the assumption when designing the layer. The authors adopted this value to ensure that no carbon deposits would be formed on the surface at these temperatures, which could affect subsequent stages of carburization.

In the case of alloy additions and their concentration in the surface layer (Figure 2), for Cr and Si, no significant differences are visible compared to the composition of the sample in the core.

In the case of the SC process, it can be noticed that Cr increases slightly near the surface, but at this stage of the research it is impossible to determine the cause. The authors plan to address this topic in future research. There is a hypothesis that this may be caused by a change in the Mn concentration in the surface layer after the SC and CTC processes, which is revealed in Figure 2. It can be noticed that from the surface into the material, the profile changes from ~0.2 wt.% to the target value in the core 1.24 wt.%. These changes can be observed up to a depth 12 µm for SC and approximately 17 µm for CTC. This means that in the subsurface layer, we deal with material strongly reduced in Mn by effusion [29], which undoubtedly results in a lower diffusion coefficient and a higher subsurface carbon content in the surface layer [30,31].

### 3.2. Microstructure and Mechanical Properties

Figure 3 shows microstructures of the surface layers after the SC and CTC processes. It can be noticed that the dominant phase structure in both cases is the structure of martensite and retained austenite, for which, in the case of the CTC process, a visibly larger amount can be observed compared to the SC process. This was confirmed by the research presented in Section 3.3.

Analyzing the size of martensite needles for particular processes, it can be noticed that for the CTC process, they are larger than for the SC process, which may indicate grain growth in the process with a constant carburizing temperature. This would justify the existence of a larger amount of austenite in the CTC process [32,33]. In the near-surface layer, it can be noticed that up to a depth of approximately 20 µm from the surface, the acicular phase and its matrix are etched more strongly, taking on a characteristic blue color, which for the Beraha II reagent may indicate the presence of ferrite and bainite in this area, instead of martensite [29,30]. Comparing it with the research results from Section 3.1, it can be concluded that the manganese effusion in this case resulted in a decrease in hardenability in this area and the critical cooling rate was not achieved in the conditions used. This is an aspect confirmed by the hardness tests presented in Figure 4.

It is noticeable that in the area affected by manganese effusion, there was a significant reduction in hardness. Because no decarburization occurred on the surface (Figure 1), the hypothesis is confirmed that the rate of cooling of the batch in this area was not effective enough for a diffusionless transformation to occur, which resulted in the appearance of bainite (Figure 3b).

Another aspect revealed by the hardness test results is lower values for the CTC process in the range of approx. 0.2–0.3 mm from the surface. Based on the results of microstructure assessment, it should be concluded that a higher content of retained austenite and manganese effusion, and therefore, different thermodynamics of the carburizing process described in Section 3.1, induced the lower mechanical properties of this area.

### 3.3. XRD Results

The retained austenite content measurements in the surface layer of gear teeth (214) after the CTC and SC processes clearly indicate that the process according to the SC recipe (215) has an advantage in this regard (Figure 5).

The average amount of retained austenite after the SC carburizing process and subsequent gas hardening at a pressure of 1.4 MPa is 13.5%, which corresponds to the content for standard low-pressure carburizing processes with gas hardening followed by tempering [32,34,35,36].

For the CTC process, the average retained austenite content is 24.4%, which confirms the observations and results presented in Section 3.2. This austenite content, apart from reducing the mechanical properties, also has a negative effect on fatigue strength [32,34,37,38,39,40,41]. Moreover, the influence of the phase structure, including the content of retained austenite discussed above, is important from the point of view of stress distribution and uniformity (Figure 6).

In the SC process, the distribution of stresses in the surface layer of the gear teeth on the pitch diameter is more even and repeatable than in the CTC process. Moreover, the reduction in the amount of retained austenite within the technological surface layer resulted in an increase in the value of compressive stresses [42,43,44,45]. This state of stress determines the SC machining as the one for which the fatigue strength will probably be characterized by higher values.

### 3.4. Dimensional Change Test Results

The tests were carried out on dimensional changes at characteristic points of the geometry, with knowledge of the results of microstructure assessment, chemical composition, stresses and retained austenite in wheels subjected to SC and CTC processes as described in Section 2.3.

The first research concerned the thickness of teeth at the pitch diameter, and the results are presented in Figure 7.

The average value of the tooth thickness change after the SC process was 0.023 mm, while in the case of the CTC process it was 0.041 mm; in both cases there was a reduction in dimension. This means that after the SC process and gas hardening, the relative deformation of the wheel teeth on the pitch diameter was reduced by approximately 45%.

It was also observed that the deformation is not uniform along the entire length of the involute (Figure 8). To confirm the formulated research thesis, a statistical analysis of the test results was conducted, based on the ANOVA test. First, it was checked if the test results obtained for particular groups could be presented as a normal distribution. On this basis, an ANOVA test was performed, which showed that for both the SC and CTC treatment values, the *P* parameter is less than 0.05. To demonstrate the significance of differences for particular groups, a Bonfferoni test was performed, which showed that when comparing head-division, head-foot and division-foot, there is a significant difference in test results (*p*-value (*t*-test) ≤ Bonfferoni correction ALPHA).

The teeth deform more strongly in the head and pitch diameter area than at the root, where the deformation is compensated by the rim. Moreover, the nature of the deformation also changes. In the area of the head and pitch diameter, the material shrank, while at the foot the tooth thickness increased. However, the nature of these deformations does not depend on the processing method.

In the case of radial tooth runout measurements, no significant differences were observed for different machining variants (Figure 9).

The value of radial runout for the wheel carburized at constant temperature and in the step carburizing option is similar, and no significant differences were observed. The maximum deviations in both cases are up to 20 µm.

However, the relative deformation of the wheel seat diameter after the pre-carburizing process (SC) is smaller than for the Constant-Temperature Carburizing (CTC) case (Figure 10). An ANOVA test, analogous to that for tooth thickness, showed the significance of differences in test results for SC and CTC machining.

In both cases, there was a reduction in the diameter of the settling hole. After pre-carburizing, the hole diameter decreased by an average of 60 µm, and after carburizing, without pre-saturation, it decreased by 76 µm.

## 4. Conclusions

Based on the measurements performed, it was found that the use of pre-carburizing at the stage of heating the steel to the target process temperature resulted in reduced deformation of gear teeth. However, it did not significantly affect the radial runout. The process of staged carburizing at different temperatures (SC process) influenced the content of retained austenite in the surface layer, which was caused, on one hand, by the grain growth in CTC process, and, on the other hand, by the effusion of manganese caused by the reduced process pressure at higher temperatures. Increasing the strength of austenite by introducing carbon at temperature stages during heating makes it possible to effectively limit tooth deformation. However, in the case of this wheel size, it has no effect on the change in radial deformation. The introduction of carbon into the steel at the heating stage causes an increase in compressive stresses in the surface layer of the steel, but the analysis of the thickness of the teeth at the foot, at the pitch diameter, and at the head showed that the nature of the deformations varies depending on the measurement point. It was shown in the manuscript that elements with developed surfaces may tend to change stresses from compressive to tensile, which may be of key importance from the point of view of the operation of such details. However, in the case of hole deformation, there is a tendency to reduce diameters.

Carbon saturation of austenite increased yield strength, and therefore the dimensional stability of steel heat-treated at elevated temperatures has also been improved, which effectively permits high-temperature treatment of critical steel parts, such as gear wheels, for which high dimensional accuracy is required.

## Figures and Tables

**Figure 1 materials-16-07630-f001:**
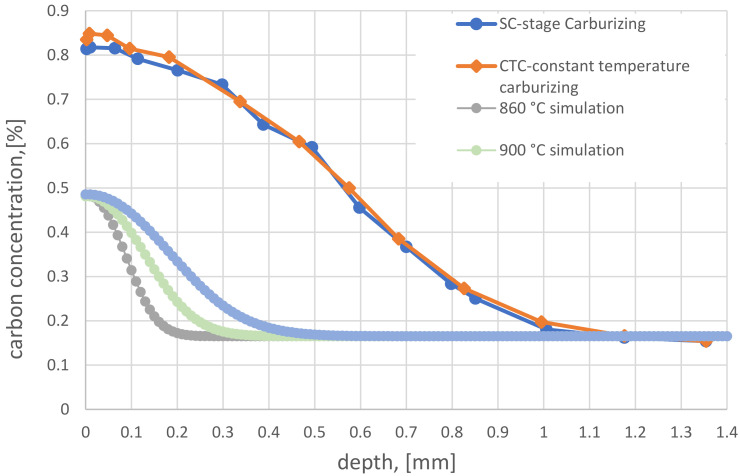
Carbon profiles for SC and CTC processes.

**Figure 2 materials-16-07630-f002:**
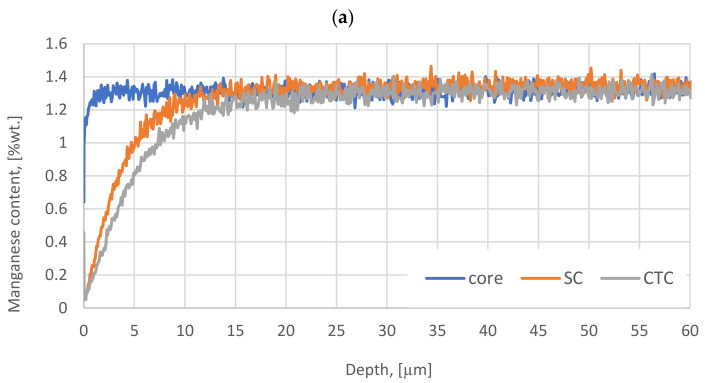
Manganese (**a**), Chromium (**b**), and Silicon (**c**) content plots on the surfaces and in the cores of wheels subjected to carburizing and case hardening.

**Figure 3 materials-16-07630-f003:**
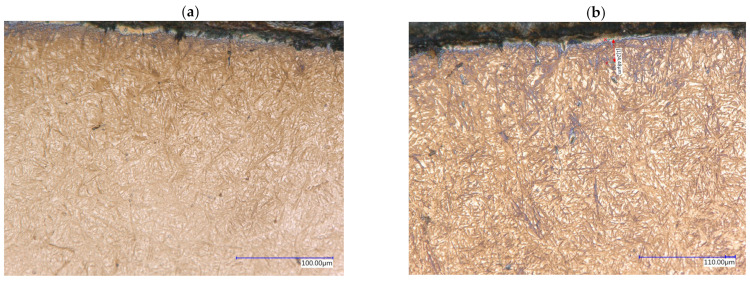
Surface layer microstructure for SC (**a**) and CTC (**b**); etchant: Beraha II.

**Figure 4 materials-16-07630-f004:**
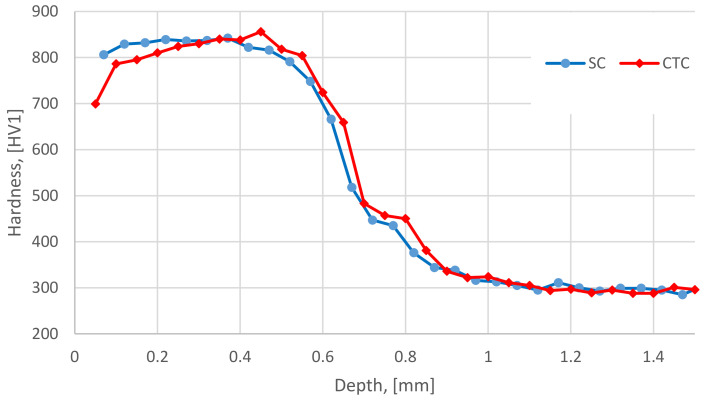
Hardness-depth dependence for SC and CTC processes.

**Figure 5 materials-16-07630-f005:**
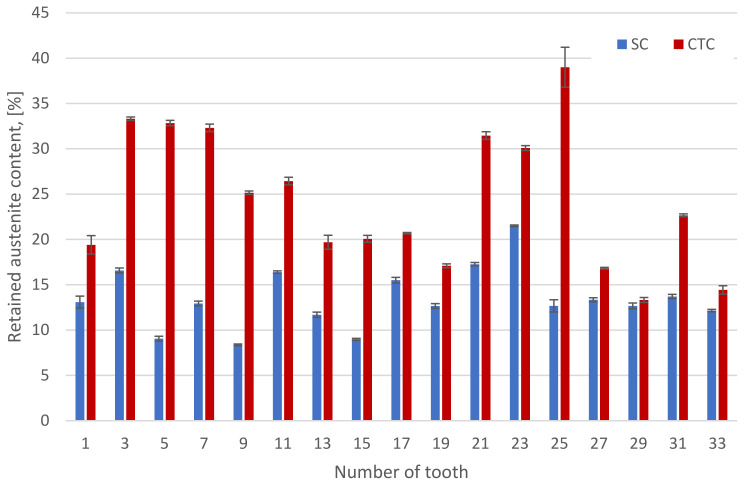
Retained austenite content in the teeth surface layers after SC and CTC treatments.

**Figure 6 materials-16-07630-f006:**
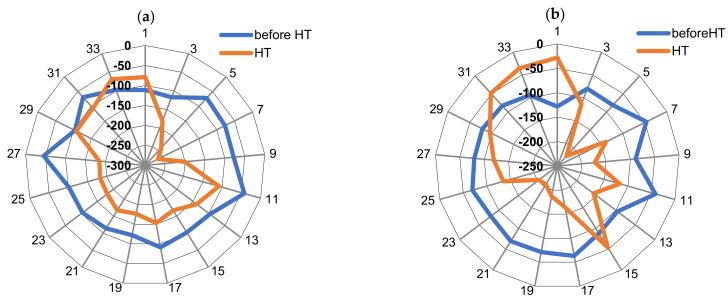
Residual stress [MPa] distribution in the teeth after SC (**a**) and CTC (**b**).

**Figure 7 materials-16-07630-f007:**
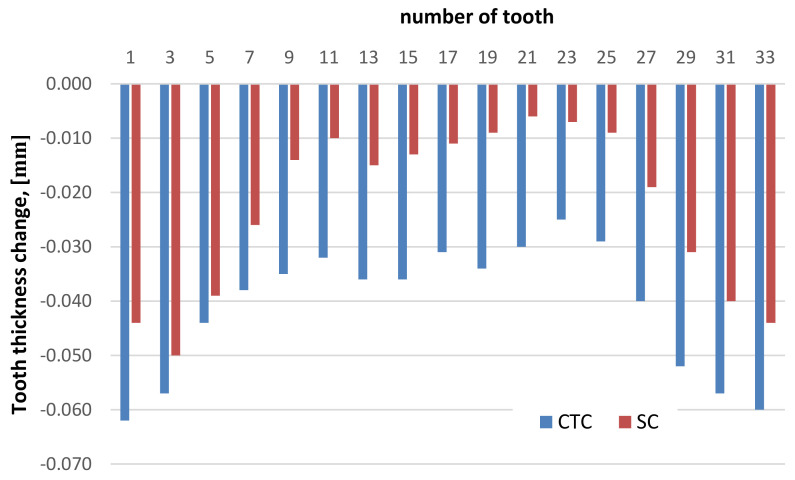
Tooth thickness change for wheels from SC and CTC processes.

**Figure 8 materials-16-07630-f008:**
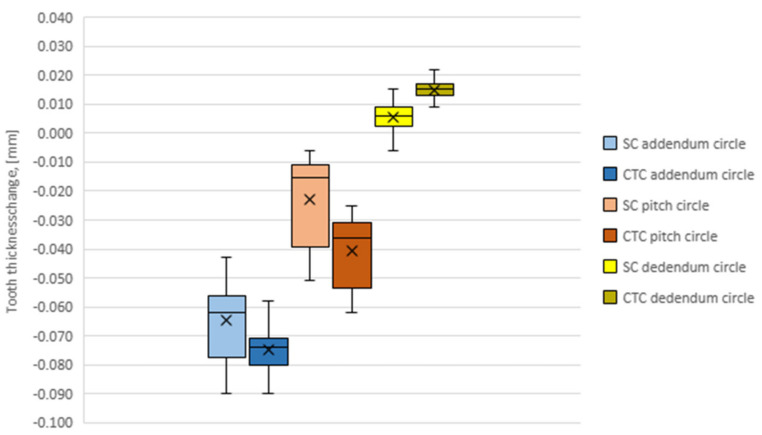
Tooth thickness change for SC and CTC processes at addendum, pitch and dedendum circles.

**Figure 9 materials-16-07630-f009:**
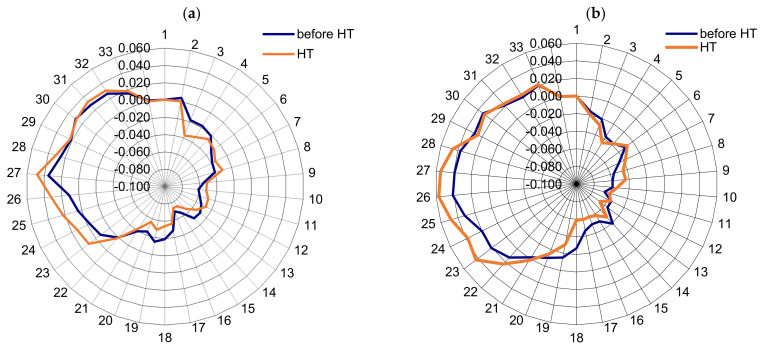
Toothing radial runout values for SC (**a**) andCTC (**b**), before and after heat treatment (HT).

**Figure 10 materials-16-07630-f010:**
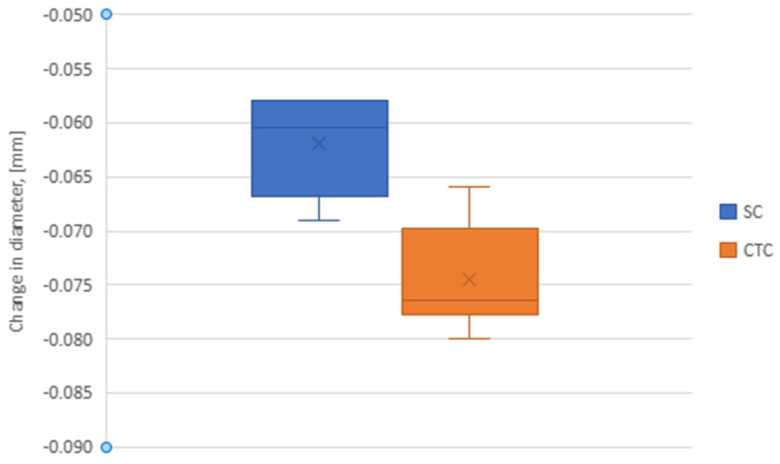
Reduction in diameter of the settling hole for SC and CTC treatments.

**Table 1 materials-16-07630-t001:** Chemical composition (wt.%) of 16MnCr5 steel.

C	Mn	Si	P	S	Cr	Ni	Mo	V	Cu	Al
0.17	1.31	0.23	0.019	0.009	0.87	0.06	0.02	0.00	0.21	0.023

**Table 2 materials-16-07630-t002:** Geometry parameters of gears tested.

Lp.	Parameter	Symbol	Value
1	Tooth number	*Z*	33
2	Nominal module	*m* _o_	5 mm
3	Gear rim width	*H*	20 mm
4	Base diameter (nominal)	*d* _a_	175 mm
5	Pitch diameter (nominal)	*D*	165 mm
6	Root diameter (nominal)	*d* _f_	153 mm
7	Mounting hole diameter	*D*	30 H7
8	Tooth thickness	*S*	7.85 mm

## Data Availability

Data are contained within the article.

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
