# Peer review of "Minimizing Deformations in High-Temperature Vacuum Carburizing"

_materials, 2023, doi:10.3390/ma16247630_

Round 1
Reviewer 1 Report
Comments and Suggestions for Authors
Introduction
· “Carburizing is one of the major surface treatments of steel. It leads to an increase of case hardness and strength while allowing the core of the steel work piece to retain its plasticity. The most recent variation of that process is carburizing carried out at reduced- pressure atmospheres known as low-pressure or vacuum carburizing. The technology has a number of advantages over carburization at endothermic atmospheres, including the possibility of performing the process at temperatures exceeding 1000ËšC [1–3], and is now replacing the latter method commonly been used before.”
Do you know the year this technology was invented? Could you include it here?
· Paragraph 2: “The extent of deformations”
How much %?
· Paragraph 2: “the difference in deformation between low-temperature and high-temperature”
How much %?
· Paragraph 3: “In extreme cases, where deformations are extensive, geometry correction may be difficult if not impossible.”
Which extreme cases?
· Please remove Figure 1 from the introduction. If you want to reference results in the introduction, use only values in text form. For example, in a previous article conducted by our research group we carried out treatment X on condition Y and found promising results of Z% and C% compared to the usual treatments...
M&M
· Pls put this sentence: “Geometry parameters of gears tested are presented in Table 2.” Together with the previous one.
Example: “Gears were made of 16MnCr5 (Table 1). After machining, the gears were vacuum annealed at a temperature 860ËšC to normalize the structure and relieve internal stresses. Geometry parameters of gears tested are presented in Table 2.”
· Centre the tables, pls.
· Be careful with the spaces after the tables.
· Be careful with the spaces before citations.
· The methodology is well written, with no further comments on this section.
R&D
· You can change this: First, the chemical composition tests checked and compared the carbon profiles obtained 149 in the step carburizing (SC) and constant temperature carburizing (CTC) processes. 150 Figure 2 shows the results of these studies.
For: First, the chemical composition tests checked and compared the carbon profiles obtained in the step carburizing (SC) and constant temperature carburizing (CTC) processes (Figure 2).
· Pls, use technical language throughout the article.
· Figure 3: Pls, reorganise the figures so that there are no blank spaces.
· Figure 4: Could you make markings (circles, arrows, etc) to show where the microstructure features you mention in the text are?
“In the near-surface layer, it can be noticed that up to a depth of approximately 20 µm” You could put a marker on this part of the figure too.
· Change: “The retained austenite content measurements in the surface layer of gear teeth after 214 the CTC and SC processes clearly indicate that the process according to the SC recipe has 215 an advantage in this regard. (Figure 6).” For: “The retained austenite content measurements in the surface layer of gear teeth after 214 the CTC and SC processes clearly indicate that the process according to the SC recipe has 215 an advantage in this regard (Figure 6).”
· Be very careful when mentioning figures dynamically, you always seem to do it wrong throughout the text.
· In relation to Figure 9, the author states that “The teeth deform more strongly in the head and pitch” However, no statistical analysis has been carried out to strengthen this claim. On the contrary, there is apparently no difference between SC and CTC in cases 1 and 2. It would be excellent to carry out an ANOVA statistical analysis to check whether or not there really is a difference. If this is done, include a section in the methodology explaining how it was done.
· Figure 10: Be careful with the quality of the figures, the rectangle around it has lines of different thickness.
· Figure 11: You can't say anything without a statistical analysis.
Comments on the Quality of English LanguageIn some cases the authors do not use the necessary technical language, as mentioned in the comments to the authors.
Author Response
Dear Reviewer,
according to revision, We have made some improvements in manuscript and we tried to answer all doubts and questions.
Below is an explanation of the specific issues:
- · “Carburizing is one of the major surface treatments of steel. It leads to an increase of case hardness and strength while allowing the core of the steel work piece to retain its plasticity. The most recent variation of that process is carburizing carried out at reduced- pressure atmospheres known as low-pressure or vacuum carburizing. The technology has a number of advantages over carburization at endothermic atmospheres, including the possibility of performing the process at temperatures exceeding 1000ËšC [1–3], and is now replacing the latter method commonly been used before.”
Do you know the year this technology was invented? Could you include it here?
We improved the abstract to make it more understandable and readable.
- Paragraph 2: “The extent of deformations” How much %?
Paragraph 2: “the difference in deformation between low-temperature and high-temperature” How much %?
Our experience, as well as the literature cited in the highlighted paragraphs, indicate that such differences and deformations increment do occur, but we do not have numerical data that would clearly indicate it in percentage terms. This is also due to the fact that the deformation values will differ, for example due to different geometries of the processed details, as well as different carburizing depths, materials and the method of cooling.
- Paragraph 3: “In extreme cases, where deformations are extensive, geometry correction may be difficult if not impossible.” Which extreme cases?
Such cases include design solutions for precision elements, where the details are ready-made before machining, with no planned allowances for grinding after the process. Vacuum carburizing, especially in combination with high-pressure cooling of the charge in gaseous media, seems to be predisposed to such solutions.
- Please remove Figure 1 from the introduction. If you want to reference results in the introduction, use only values in text form. For example, in a previous article conducted by our research group we carried out treatment X on condition Y and found promising results of Z% and C% compared to the usual treatments...
According to your suggestions, appropriate changes in the manuscript have been provided.
- Pls put this sentence: “Geometry parameters of gears tested are presented in Table 2.” Together with the previous one.
Example: “Gears were made of 16MnCr5 (Table 1). After machining, the gears were vacuum annealed at a temperature 860ËšC to normalize the structure and relieve internal stresses. Geometry parameters of gears tested are presented in Table 2.”
According to your suggestions, appropriate changes in the manuscript have been provided.
- Centre the tables, pls.
Be careful with the spaces after the tables.
Be careful with the spaces before citations.
According to your suggestions and comments, appropriate changes in the manuscript have been provided.
- You can change this: First, the chemical composition tests checked and compared the carbon profiles obtained 149 in the step carburizing (SC) and constant temperature carburizing (CTC) processes. 150 Figure 2 shows the results of these studies.
For: First, the chemical composition tests checked and compared the carbon profiles obtained in the step carburizing (SC) and constant temperature carburizing (CTC) processes (Figure 2).
According to your suggestions, appropriate changes in the manuscript have been provided.
- Figure 3: Pls, reorganise the figures so that there are no blank spaces.
According to your suggestions, appropriate changes in the manuscript have been provided.
- “In the near-surface layer, it can be noticed that up to a depth of approximately 20 µm” You could put a marker on this part of the figure too.
Figure 4: Could you make markings (circles, arrows, etc) to show where the microstructure features you mention in the text are?
According to your suggestion, I marked an area 20µm from the surface. Unfortunately, I don't quite understand what I should mark with arrows or circles.
- Change: “The retained austenite content measurements in the surface layer of gear teeth after 214 the CTC and SC processes clearly indicate that the process according to the SC recipe has 215 an advantage in this regard. (Figure 6).” For: “The retained austenite content measurements in the surface layer of gear teeth after 214 the CTC and SC processes clearly indicate that the process according to the SC recipe has 215 an advantage in this regard (Figure 6).”
According to your suggestions, appropriate changes in the manuscript have been provided.
- In relation to Figure 9, the author states that “The teeth deform more strongly in the head and pitch” However, no statistical analysis has been carried out to strengthen this claim. On the contrary, there is apparently no difference between SC and CTC in cases 1 and 2. It would be excellent to carry out an ANOVA statistical analysis to check whether or not there really is a difference. If this is done, include a section in the methodology explaining how it was done.
I performed an ANOVA (Analysis of Variance) and the Bonfferoni test, which showed that for each group taken for comparison, i.e. addendum and pitch diameter, addendum and dedendum circle, and pitch diameter and addendum of the teeth, the analysis showed significance of differences in values (p crit<0.05 ). However, the analysis also showed that the values of dimensional changes for the head and the pitch diameter are the largest, but I agree that it does not depend on the method of processing (SC or CTC). SC processing was characterized by smaller relative values of dimensional changes in each variant. In the manuscript, I changed the graphical form of Figure 9 to clarify and highlight results.
- Figure 10: Be careful with the quality of the figures, the rectangle around it has lines of different thickness.
According to your suggestions, appropriate changes in the manuscript have been provided.
- Figure 11: You can't say anything without a statistical analysis.
The same analysis of variance was performed as described in point 11. The obtained analysis results indicate that there is a significant difference in values between the SC and CTC processing (pcrit<0.05). To better illustrate the research results, Figure 11 was changed to a box plot.
Reviewer 2 Report
Comments and Suggestions for Authors
In this work, the authors studied the reducing deformation resulting from high-temperature vacuum carburizing and post carburizing heat treatment. This is in general a good work, and the reviewer believes that it can be accepted after moderation revisions.
Figure 1 xlabel, the unit of temperature needs to be fixed.
Table 1, the Mn is not in bold font while the others are bold.
Many format and typo issues are detected, please check the whole manuscript again and fix them.
The annotation in Figure 2 needs to be fixed. For example, 860degC, this is not professional.
Figure 3, xlabel, should be in μm not um.
What is the unit of the value in Figure 7?
some of the figures have a whole framework some not.
Comments on the Quality of English Language
Moderate modification needed.
Author Response
Dear Reviewer,
according to revision, We have made some improvements in manuscript and we tried to answer all doubts and questions.
Below is an explanation of the specific issues:
- Figure 1 xlabel, the unit of temperature needs to be fixed.
This Figure has been removed from the manuscript.
- Table 1, the Mn is not in bold font while the others are bold.
According to your suggestions and comments, appropriate changes have been provided.
- Many format and typo issues are detected, please check the whole manuscript again and fix them.
According to your suggestions and comments, appropriate changes in the manuscript have been provided.
- The annotation in Figure 2 needs to be fixed. For example, 860degC, this is not professional.
According to your suggestions and comments, appropriate changes have been provided.
- Figure 3, xlabel, should be in μm not um.
According to your suggestions and comments, appropriate changes have been provided.
- What is the unit of the value in Figure 7?
According to your suggestions and comments, appropriate changes in description have been provided.
- some of the figures have a whole framework some not.
According to your suggestions and comments, appropriate changes have been provided.
Round 2
Reviewer 2 Report
Comments and Suggestions for Authors
I believe this work can be accepted after the authors' revision.